# Children with Chronic Immune Thrombocytopenia Exhibit High Expression of Human Endogenous Retroviruses TRIM28 and SETDB1

**DOI:** 10.3390/genes14081569

**Published:** 2023-08-01

**Authors:** Pier-Angelo Tovo, Ilaria Galliano, Emilia Parodi, Cristina Calvi, Stefano Gambarino, Francesco Licciardi, Maddalena Dini, Paola Montanari, Margherita Branca, Ugo Ramenghi, Massimiliano Bergallo

**Affiliations:** 1Department of Public Health and Pediatric Sciences, University of Turin, Piazza Polonia 94, 10126 Turin, Italy; pierangelo.tovo@unito.it (P.-A.T.); ugo.ramenghi@unito.it (U.R.); 2Pediatric Laboratory, Department of Public Health and Pediatric Sciences, University of Turin, Regina Margherita Children’s Hospitalno, Piazza Polonia 94, 10126 Turin, Italy; ilaria.galliano@unito.it (I.G.); cristina.calvi@unito.it (C.C.); gambarino.stefano@gmail.com (S.G.); maddalena.dini@edu.unito.it (M.D.); paola.montanari@unito.it (P.M.); 3Pediatric and Neonatology Unit, Ordine Mauriziano Hospital, Largo Filippo Turati 62, 10128 Turin, Italy; emilia.parodi@unito.it; 4Regina Margherita Children’s Hospital, Piazza Polonia 94, 10126 Turin, Italy; francesco.licciardi@gmail.com; 5Postgraduate School of Pediatrics, University of Turin, Piazza Polonia 94, 10126 Turin, Italy; margherita.branca@edu.unito.it

**Keywords:** autoimmune disease, primary immune thrombocytopenia, endogenous retroviruses, SETDB1, TRIM28, genetic pathomechanism, children

## Abstract

Chronic immune thrombocytopenia (CITP) is an autoimmune disease whose underlying biologic mechanisms remain elusive. Human endogenous retroviruses (HERVs) derive from ancestral infections and constitute about 8% of our genome. A wealth of clinical and experimental studies highlights their pivotal pathogenetic role in autoimmune diseases. Epigenetic mechanisms, such as those modulated by TRIM28 and SETDB1, are involved in HERV activation and regulation of immune response. We assessed, through a polymerase chain reaction real-time Taqman amplification assay, the transcription levels of pol genes of HERV-H, HERV-K, and HERV-W; env genes of Syncytin (SYN)1, SYN2, and HERV-W; as well as TRIM28 and SETDB1 in whole blood from 34 children with CITP and age-matched healthy controls (HC). The transcriptional levels of all HERV sequences, with the exception of HERV-W-env, were significantly enhanced in children with CITP as compared to HC. Patients on eltrombopag treatment exhibited lower expression of SYN1, SYN2, and HERV-W-env as compared to untreated patients. The mRNA concentrations of TRIM28 and SETDB1 were significantly higher and were positively correlated with those of HERVs in CITP patients. The over-expressions of HERVs and TRIM28/SETDB1 and their positive correlations in patients with CITP are suggestive clues of their contribution to the pathogenesis of the disease and support innovative interventions to inhibit HERV and TRIM28/SETDB1 expressions in patients unresponsive to standard therapies.

## 1. Introduction

Immune thrombocytopenia (ITP), also called idiopathic or immune thrombocytopenic purpura, is a bleeding disorder characterized by isolated thrombocytopenia with a peripheral blood platelet count < 100 × 10^9^/L [1]. It is the most common hematological disease in childhood [2]. Primary ITP must be distinguished from secondary forms, which are triggered by an underlying disease, such as infections, autoimmune disorders, and drug exposures [3]. Children with ITP are usually well appearing, with only bleeding signs, often with sudden onset and a negative family and past medical history. Primary ITP is often benign and self-limited in childhood, though in 20–30% of cases it may give rise to chronic forms lasting beyond 12 months (CITP, chronic immune thrombocytopenia) [1,4]. Older age, higher initial platelet count, and female gender may be predictive elements for chronic evolution [5,6]. 

ITP is considered an immune-mediated disease and platelet autoantibodies (PAA) are thought to be important actors [7]. Testing for PAA is, however, not recommended for the diagnosis, due to its poor accuracy. Growing evidence highlights that more complex immune mechanisms, such as T cell-induced effects, are implicated in ITP pathogenesis [8,9]. In particular, loss of tolerance and exposure to infection-driven post-translational changes of platelet antigens, promoting T cell-conditioned B cell responses, or T cells cross-reactive against platelets and pathogen-derived peptides have been suggested [10]. The essential role played by micro-RNAs has also been proposed [11]. Dysregulation of epigenetic processes leading to abnormal gene expression via variations in the DNA methylation and in histone-induced heterochromatin formation, has become an attractive hypothesis too [12,13].

Human endogenous retroviruses (HERVs) originate from ancestral infections of somatic cells with subsequent integration into germ line of primates millions of years ago. During evolution, the continuous mutations have blocked the production of infectious virions. However, HERVs maintain their retroviral structure with three principal genes: group associated antigens (*gagitalics is not necessary, but it must be consistent in the whole papper*), polymerase (*pol*), and envelope (*env*), flanked between two regulatory long terminal repeats (LTRs) [14]. Most HERVs are inactive, but some sequences are transcribed and a few encode proteins. An increasing body of literature heralds the crucial role of HERVs during the intrauterine life [15,16], representing a clear example of how retroviral elements are used by the host for its own survival. For instance, two envelope proteins, called syncytin-1 (SYN1) [17] and syncytin-2 (SYN2) [18], are essential for the placental morphogenesis and contribute to the materno-fetal tolerance [19,20]. Postnatally, the physiologic functions of HERVs remain questionable, whereas their abnormal expressions may induce several hallmarks of cells infected by a virus, leading to innate and adaptive immune responses. HERV alterations have been associated with an increasing number of diseases, including autoimmune disturbances [21,22,23,24]. Actually, HERVs can regulate the expression of close cellular genes [25,26]. Their RNAs, through a copy and paste mechanism, can be reintegrated into the DNA or be recognized as non-self by specific viral receptors, triggering a variety of inflammatory and immune reactions [22,25,26,27,28]. Furthermore, HERV antigens can induce targeted immune responses, including specific and/or cross-reactive antibodies with tissue epitopes [29,30,31,32]. The persistence of aberrant HERV expressions may vary over time [33,34]; their dysregulation can thus determine an episodic or a chronic challenge for the immune system [21], reminiscent of the clinical presentation of ITP.

HERV expression is modulated by environmental factors via epigenetic mechanisms. Tripartite motif containing 28 (TRIM28), also called KAP1 or TIF1-β, is a nuclear co-repressor of Krüppel-associated box domain zinc finger proteins (KRAB-ZFPs) [35,36]. SET domain bifurcated histone lysine methyltransferase 1 (SETDB1), also referred to as ESET, is a methyltransferase with high specificity for the lysine 9 residue of histone H3 [37]. It contributes to orchestrate the heterochromatin formation. Both TRIM28 and SETBD1 play key roles for epigenetic transcriptional modulation of retroviral sequences [38,39]. Furthermore, accumulating data demonstrate their direct involvement in cellular homeostasis and in the control of both innate and adaptive immune responses [40,41]. 

Despite the wealth of data supporting the role of HERVs, TRIM28, and SETDB1 in eliciting and/or maintaining autoimmune diseases, no studies explored their expressions in patients affected by ITP. Therefore, the aims of this study were to assess and compare the transcriptional levels of pol genes of HERV-H, -K, and -W, the three retroviral families most widely studied [19,21]; env genes of SYN1, SYN2, and HERV-W [17,18,22]; as well as TRIM28 and SETDB1 in whole blood from children affected by CITP and in age-matched healthy controls (HC).

## 2. Material and Methods

### 2.1. Study Populations

All children and adolescents affected by CITP (i.e., a primary immune thrombocytopenia lasting for more than 12 months according to the IWG criteria [1]) and regularly followed at the Department of Pediatrics, University of Turin, Regina Margherita Children’s Hospital, Turin, Italy, were enrolled in the study. Patients with secondary forms of thrombocytopenia, associated hematologic/immunologic abnormalities, and/or comorbidities were excluded from the analysis.

Asymptomatic subjects of comparable age who were tested at the same hospital for routine laboratory examinations and whose results were all within the normal reference range were the control group. Subjects with any confirmed or suspected disease, such as infections, cancer, autoimmune disorders, inflammatory diseases, neurological disturbances, or abnormal laboratory results were excluded from the study. 

### 2.2. RNA Extraction and Retrotranscription

Total RNA was extracted from whole blood with RNA Blood Kit (Maxwell automated extractor, Promega, Madison, WI, USA). This kit provides treatment with DNase during the first step of RNA extraction. To exclude contamination with residual genomic DNA, total RNA extracts were directly amplified without reverse transcription. RNA concentration and purity were spectroscopy determined (ND-1000 spectrophotometer, Biochrom Enterprise, Waterbeach, Cambridge, UK) at absorbance of 260 and 280 nm. RNAs were stored at −80 °C until use.

A total of 400 ng of RNA was reverse-transcribed with 2 μL of buffer 10× 4.8 μL of MgCl_2_ 25 mM, 1 μL of RNase inhibitor 20 U/l, 2 μL ImpromII (Promega), 2 μL mix dNTPs 100 mM (Promega), 0.4 μL random hexamers 250 μM (Promega), and dd-water in a final volume of 20 μL. The reaction mix was carried out in a GeneAmp PCR system, 9700 Thermal Cycle (Applied Biosystems, Foster City, CA, USA) under the following conditions: 5 min at 25 °C, 60 min at 42 °C, and 15 min at 70 °C for the inactivation of enzymes; the cDNAs were stored at −20 °C until use.

### 2.3. Transcription Levels of Pol Genes of HERV-H, -K, and -W; Env Genes of SYN1, SYN2, and HERV-W; as well as of TRIM28 and SETB1 by Real-Time PCR Assay

Transcription levels of pol genes of HERV-H, HERV-K, and HERV-W; env genes of SYN1, SYN2, and HWERV-W; and TRIM28 and SETDB1 were achieved as previously described in detail [23,24,33,34] using the primers and probes reported in Table 1. Briefly, 40 ng of cDNA was amplified in a 20 μL total volume reaction, containing 2.5 U goTaQ MasterMix (Promega), 1.25 mmol/L MgCl_2_, 200 nmol of specific probes, and 500 nmol of specific primers.

All amplifications were performed in a 96-well plate at 95 °C for 10 min, followed by 45 cycles at 95 °C for 15 s and at 60 °C for 1 min. Each sample was run in triplicate. Relative quantification of target gene transcripts was performed using the 2^−ΔΔCt^ method [42]. GAPDH was selected as the reference gene because it has been shown to have good efficiency and excellent reproducibility with constant expression in human leukocyte samples and has been used previously in our studies [23,24,33,34]. Briefly, after normalizing the PCR result of each target gene with the GAPDH, the method includes an additional calibration of this value with the median expression of the same gene obtained in a pool of healthy controls after normalization with the GAPDH. The results, expressed in arbitrary units (RQ), show the variations in target gene expression compared with standard controls. Because we determined the cycle threshold (Ct) for each target in all samples, we argue that our methods were suitable for HERVs, TRIM28, and SETDB1 detection and quantifications. 

### 2.4. Statistical Analysis

One-way ANOVA test was used to compare the age between patients with CITP and the two subgroups (B1 and B2) of HC. Mann–Whitney test was used to compare the transcripts of pol genes of every HERV family; env genes of SYN1, SYN2, and HERV-W; as well as TRIM28 and SETDB1 between children with CITP and control children. Spearman correlation test was performed to estimate the correlations concerning transcription levels of each HERV sequence and TRIM28 or SETDB1 in every group of children. Statistical analyses were performed using Prism software, version 7 (GraphPad Software, La Jolla, CA, USA). In all analyses, *p* < 0.05 was taken to be statistically significant.

## 3. Results

### 3.1. Study Populations

As reported in Table 2, 34 children with CITP were enrolled in the study (Group A). Healthy controls (HC) were subdivided into two subgroups: Group B1 included 64 children and adolescents who had been enrolled as healthy controls in our previous studies on pol gene expressions of HERV-H, -K, and -W [23,24,33,34]; Group B2 included 47 subjects who were tested for expression of env genes of SYN1, SYN2, and HERV-W, and of TRIM28 and SETDB1. The median ages were comparable among the three groups of children (one-way ANOVA, *p* = 0.1998). 

### 3.2. Expression Levels of Housekeeping Gene

The transcription levels of housekeeping gene GAPDH were similar between patients with CITP and HC: Medians and IQR 25%–75%: Group A: median 22.3, IQR 21.8–22.7; Group B1: median 22.1, IQR 21.6–22.5 (*p* = 0.4111); Group B2: median 21.8, IQR 21.5–22.4 (*p* = 0.1432).

### 3.3. Expressions of HERV-H-pol, HERV-K-pol, HERV-W-pol, SYN1-env, SYN2-env, and HERV-W-env

The pol genes of HERV-H, HERV-K, and HERV-W and the env genes of SYN1, SYN2, and HERV-W were transcriptionally active in all the study populations. 

Figure 1 illustrates that the medians of the transcription levels of HERV-H-pol, HERV-K-pol, and HERV-W-pol were significantly higher in patients with CITP (Group A) as compared to 64 healthy controls (Group B1). The same figure also shows that the medians of the mRNA levels of env genes of SYN1 and SYN2 were significantly higher in patients with CITP (Group A) as compared with 47 healthy controls (Group B2), while no statistically significant difference was found for HERV-W-env between the two groups.

Medians and IQR 25%–75%: HERV-H-pol: Group A median 1.44, IQR 1.20–1.78; Group B1 median 1.02, IQR 0.81–1.27; HERV-K-pol: Group A median 1.74, IQR 1.24–1.96; Group B1 median 1.01, IQR 0.87–1.21; HERV-W-pol: Group A median 1.80, IQR 1.46–2.24; Group B1 median 1.09, IQR 0.73–1.53; SYN1-env: Group A median 2.61, IQR 2.21–3.93; Group B2 median 1.01, IQR 0.81–1.24; SYN2-env: Group A median 1.50, IQR 1.10–1.99; Group B2 median 0.98, IQR 0.77–1.16; HERV-W-env: Group A median 1.07, IQR 0.92–1.49; Group B2 median 1.00, IQR 0.81–1.26. 

The values of each HERV RNA quantification for individual patients are reported in Appendix A.

### 3.4. HERV Expressions in Patients with and without Eltrombopag Treatment

As illustrated in Figure 2, the expressions of SYN1 and SYN2 were inhibited significantly, with borderline *p* value for HERV-W-env, in the 11 patients on eltrombopag treatment alone as compared to the 18 patients without any treatment (including no administration of steroids or IVIgG). When expressions of HERV-env genes from patients on eltrombopag alone were compared to those from HC, only SYN1 showed significantly higher values (Appendix A). No differences were found for HERV-pol genes between patients on eltrombopag treatment alone and those with no therapy. Medians and IQR 25%–75%: SYN1: on eltrombopag (E-pag) median 2.48, IQR 1.97–2.97; no treatment (No-T) median 3.48, IQR 2.44–4.71; SYN2: E-pag median 1.14, IQR 1.01–1.49; No-T median 1.80, IQR 1.34–2.12; HERV-W-env: E-pag median 0.95, IQR 0.85–1.28; No-T median 1.36, IQR 1.02–1.64; HERV-H-pol: E-pag median 1.44, IQR 1.04–2.05; No-T median 1.60, IQR 1.34–1.78; HERV-K-pol: E-pag median 1.73, IQR 1.18–1.97; No-T median 1.77, IQR 1.41–2.03; HERV-W-pol: E-pag median 1.73, IQR 1.31–2.22; No-T median 1.87, IQR 1.56–2.24.

### 3.5. Expressions of TRIM28 and SETDB1

As detailed in Figure 3, the transcriptional levels of TRIM28 and SETDB1 were significantly higher in patients with CITP (Group A) than HC (Group B2). Medians and IQR 25%–75%: TRIM28: Group A median 1.56, IQR 1.21–1.96; Group B2 median 1.00, IQR 0.79–1.30; SETDB1: Group A median 1.56, IQR 1.29–2.00; Group B2 median 1.00, IQR 0.84–1.20. 

The mRNA values of TRIM28 and SETDB1 for individual patients are reported in Appendix A.

### 3.6. Correlations between Expressions of TRIM28 and HERVs

The mRNA levels of all HERV sequences positively correlated with the mRNA levels of TRIM28 in patients with CITP (Figure 4), whereas no significant correlations were found in HC (Appendix A).

### 3.7. Correlations between Expressions of SETDB1 and HERVs

The transcription levels of all HERV sequences were positively correlated with those of SETDB1 in patients with CITP (Figure 5); whereas, no significant correlations were found in healthy children (Appendix A).

### 3.8. Expressions of TRIM28 and SETDB1 in Patients with and without Eltrombopag Treatment

The median transcriptional levels of TRIM28 and SETDB1 were lower in the 11 patients on eltrombopag treatment alone as compared to the 18 patients with no treatment, but they did not reach statistically significant differences, with borderline *p* value for TRIM28 (Appendix A). Medians and IQR 25%–75% of TRIM28: on eltrombopag: median 1.26, IQR 0.99–1.76; no treatment: median 1.75, IQR 1.37–1.99 (*p* = 0.0682). Medians and IQR 25%–75% of SETDB1: on eltrombopag: median 1.64, IQR 1.27–1.80; no treatment: median 1.80, IQR 1.46–2.17 (*p* = 0.3397). 

## 4. Discussion

Current results highlight for the first time that children and adolescents with CITP exhibit higher transcriptional levels of pol genes of HERV-H, -K, and -W and env genes of SYN1 and SYN2 in whole blood as compared to age-matched healthy subjects. Some clues that endogenous retroviruses could be involved in inducing alterations in platelet count can be found in the literature. Retrovirus-like particles were detected in lysates of purified platelets [43,44]. Reverse transcriptase activity and HERV-K-pol-related RNAs were found at high levels in platelets of patients with essential thrombocythemia [45]. A HERV-K-gag protein was evidenced in the cytoplasm and viral particles in the cell membrane and into vacuoles of megakaryocytes from two patients with essential thrombocythemia; whereas, none of these elements were found in megakaryocytes from a normal subject [46]. Furthermore, non-autoimmune mice developed microvascular platelet aggregation and thrombocytopenia upon transplantation of a hybridoma clone secreting monoclonal antibodies reactive with an endogenous retroviral env component [47]. 

The cause and the clinical relevance of the enhanced expression of HERV sequences in CITP children remain elusive. Several experimental studies demonstrate that defective transcription of TRIM28 and SETDB1 may lead to increased HERV activation, via DNA hypomethylation and limited chromatin condensation [38,39]. The mRNA levels of TRIM28 and SETDB1 were, however, not compromised, but increased in our patients, suggesting that the HERV overexpression cannot be ascribed to reduced transcription of TRIM28 or SETDB1. Interestingly enough, no impaired TRIM28/SETDB1 transcriptions were also observed in other settings of children with immune-mediated diseases characterized by upregulation of HERV sequences [33,34,48]. To this purpose, it must be underlined that the protein complex TRIM28/SETDB1 is crucial for keeping endogenous retroviruses in a silent state in pluripotent stem cells and early embryos [49,50]. In contrast, when these cells differentiate into various somatic cells, transcription of retroviral elements becomes independent of such repressors [51], which may also act as transcriptional activators rather than as repressors [52,53]. This might be the case in CITP patients too. In this context, it is worth noting the positive correlations found between mRNA levels of TRIM28 or SETDB1 and HERVs. This suggests that the former might play relevant regulatory functions on the latter. Upstream regulatory pathways could, however, justify the simultaneously increased expression of cellular genes, such as TRIM28/SETDB1, and retroviral sequences. Furthermore, their functional interactions might derive from post-translational conditions between encoded proteins, while we evaluated only their transcriptional landscape. 

Viral infections often precede the onset of primary ITP by a few weeks. The virus-driven stimulation of the immune system is thought to be highly implicated in inducing targeted injuries [54,55]. Several viruses, such as herpesviruses [56,57], HIV [58], HCV [33], influenza type A [59], and SARS-CoV-2 [34,60] can trigger HERV transactivation. Exogenous viral infections elicit an inflammatory response via the TLR/NF-kB pathway resulting in the synthesis of interferons (IFNs) and pro-inflammatory cytokines. After passage into the nucleus, NF-kB binds to target elements of HERV proviruses and, with the synergistic action of IFNs and inflammatory cytokines, enhances HERV transcriptions [61]. The HERV expression increases the cell resistance to secondary infections [27], but it exposes the cell to the potential negative effects of endogenous retroviruses. Actually, HERVs can alter the transcription of adjacent cellular genes [25,26]. Their mRNAs can be retro-transcribed into DNA causing significant mutations of cellular genes or, if being sensed as non-self by viral receptors, may induce inflammatory and immune reactions [14]. Furthermore, retroviral epitopes can trigger specific antibodies or share molecular mimicry with a few body tissues [29,30,31,32]. The final result may be an autoimmune-mediated destruction and a defective production of platelets [62]. Actually, a wealth of clinical [21,23,24] and experimental findings [22,63,64] supports the pathogenetic role of HERVs in autoimmune diseases. HERV-encoded proteins are targets of specific reaction in patients with SLE [32]. Higher quantities of surface HERV-H and HERV-W env proteins with specific seroreactivity were found in leukocytes from patients affected by multiple sclerosis [29]. A significant up-regulation in HERV-K mRNA concentrations [65] and enhanced antibody response to a HERV-K-gag [30] and -env [31] peptides were observed in patients with rheumatoid arthritis, while a HERV-K superantigen was detected in those with juvenile arthritis [66].

The number of HERVs with high mRNA levels in CITP patients suggests that their upregulation is a wide and complex phenomenon. As regards the single HERV genes, pol genes of HERV-H, HERV-K, and HERV-W contribute to the production of reverse polymerase, that catalyzes the mRNA retro-transcription into the DNA. This may result in new mutations or dysregulated transcription of cellular genes. It must be underlined that the primers and probes we used revealed preserved sequences of many pol genes from different HERV families, which, however, did not allow outlining of the effects of single HERV loci. SYN1 and SYN2 share syncytial and immunomodulatory properties. They target T-cell activation and significantly suppress T-cell functions [19,20]. SYN1 is usually detectable in all circulating leucocytes; upon stimulation it promotes rapid activation of monocytes [19] and release of chemokines and cytokines [67,68]. Taken together, all these findings support a critical role of HERV over-expressions in CITP. However, as for other immuno-mediated diseases, whether their activation is a cause or an effect of autoimmune reactions remains an unresolved enigma. 

Eltrombopag is a nonpeptide thrombopoietin receptor agonist used for treatment of CITP as a highly effective and well-tolerated therapy for achieving hemostatic platelet counts [69,70]. A third of our patients were on eltrombopag therapy alone. Interestingly enough, these patients exhibited significantly impaired transcriptions of SYN1, SYN2, and HERV-W-env as compared to untreated patients. One wonders whether this downregulation of HERV-env sequences may contribute, at least in part, to the positive effects of eltrombopag treatment in patients with CITP. 

Epigenetic processes encompass chromatin-based mechanisms in the regulation of gene expression with no structural DNA changes. Environmental factors can cause epigenetic alterations, such as those induced by TRIM28 and SETDB1. TRIM28, through binding to lysine residues of target proteins, causes their phosphorylation and proteasome-driven degradation. TRIM28 recruits SETDB1 for SUMOylation, a post-translational event involved in several cell functions, such as protein degradation, RNA splicing, and transcriptional inhibition [71,72]. TRIM28 and SETDB1 are implicated in a multitude of epigenetic processes [72] and exert relevant actions on both innate and adaptive immune responses [39,40,41]. They influence B cell lineage differentiation and T cell lineage commitment and activation [73,74]. In particular, through the complex with KRAB-ZFPs and forkhead box P3 (Foxp3), TRIM28/SETDB1 control the expansion of T cells into regulatory phenotypes [40,73], conditioning the Treg suppressor activity or distribution of DCs and T cell priming toward inflammatory effector cells [73,75,76]. Notably, Foxp3 gene polymorphisms are also candidate risk factors in autoimmune diseases, including CITP [1,9]. 

## 5. Conclusions

Better understanding of the physiopathology of ITP and the failure of tolerance checkpoints responsible for autoantibody formation may open the way to new diagnostic and prognostic approaches and innovative therapeutic strategies. For instance, early detection and longitudinal analysis of the variables here taken into consideration might predict acute or chronic evolution of ITP and allow a patient-tailored therapy. A number of anti-HERV therapeutic strategies might be adopted, such as specific anti-RNAs, monoclonal antibodies, cytotoxic T lymphocytes against HERV antigens, and antiretroviral therapies [77,78,79]. HIV+ subjects are treated for years with cocktails of antiretroviral drugs; their optimal doses and side effects are well-known also in children. In HIV+ subjects, antiretroviral therapy blocks the activation not only of HIV, but also of HERVs [80,81], and resolution of thrombocytopenia has been noticed in coincidence with a specific treatment in a seropositive patient [82]. A combined use of antiretrovirals for six months in patients with amyotrophic lateral sclerosis to block the HERV-K hyperexpression showed an apparent better disease course in those with positive antiviral results [83]. The downregulation of syncytins and HERV-W-env in patients on eltrombopag treatment, if confirmed by targeted studies, may pave the way for its use in patients in whom enhanced activation of HERV-env sequences is thought to be responsible for their pathology. Accumulating evidence highlights that dysregulations in epigenetic mechanisms are involved in the development of CITP. Abnormal epigenetic changes can be targeted by specific small molecule compounds [84]. Therefore, given the strict interactions of HERVs and TRIM28/SETDB1 with the immune system, our results suggest that they might be implicated the physiopathology of CITP and support innovative therapeutic interventions to inhibit their expression in patients affected by CITP unresponsive to standard therapy.

## Figures and Tables

**Figure 1 genes-14-01569-f001:**
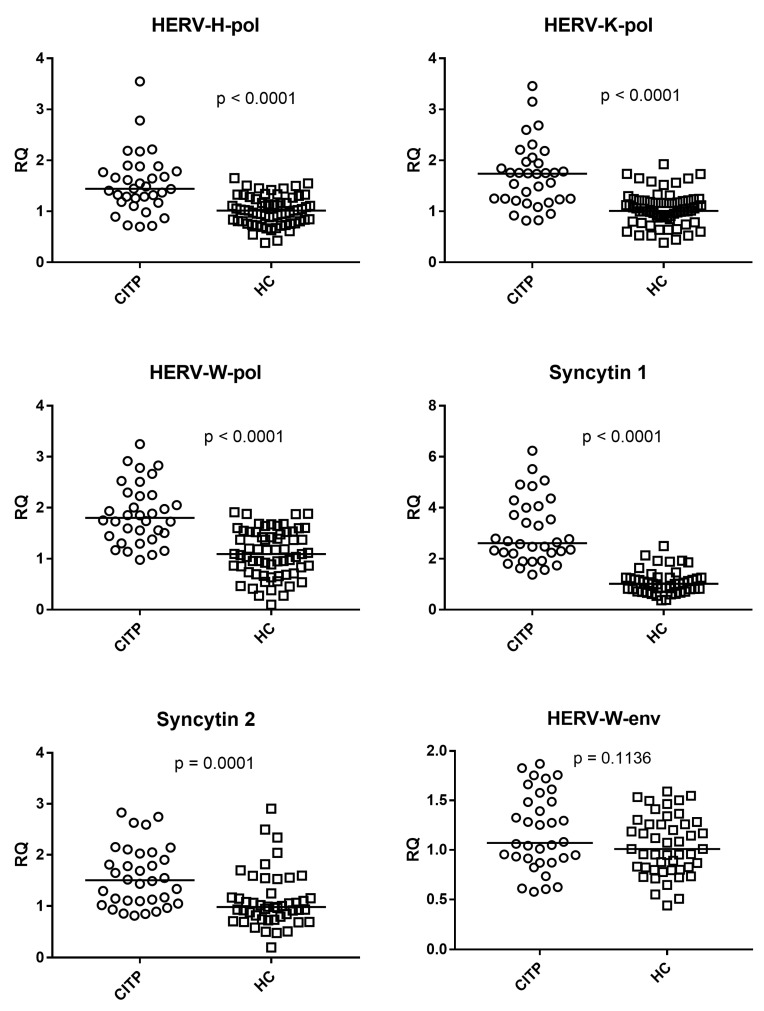
Transcription levels of pol genes of HERV-H, HERV-K, HERV-W and of env genes of Syncytin 1, Syncytin 2, and HERV-W in whole blood from patients with chronic immune thrombocytopenia (CITP) and healthy controls (HC). RQ: Relative Quantification. Circles and squares show the median of three individual measurements, horizontal lines are the median values. Statistical analysis: Mann–Whitney test was used to compare the transcriptional levels of each target gene between children with CITP and HC.

**Figure 2 genes-14-01569-f002:**
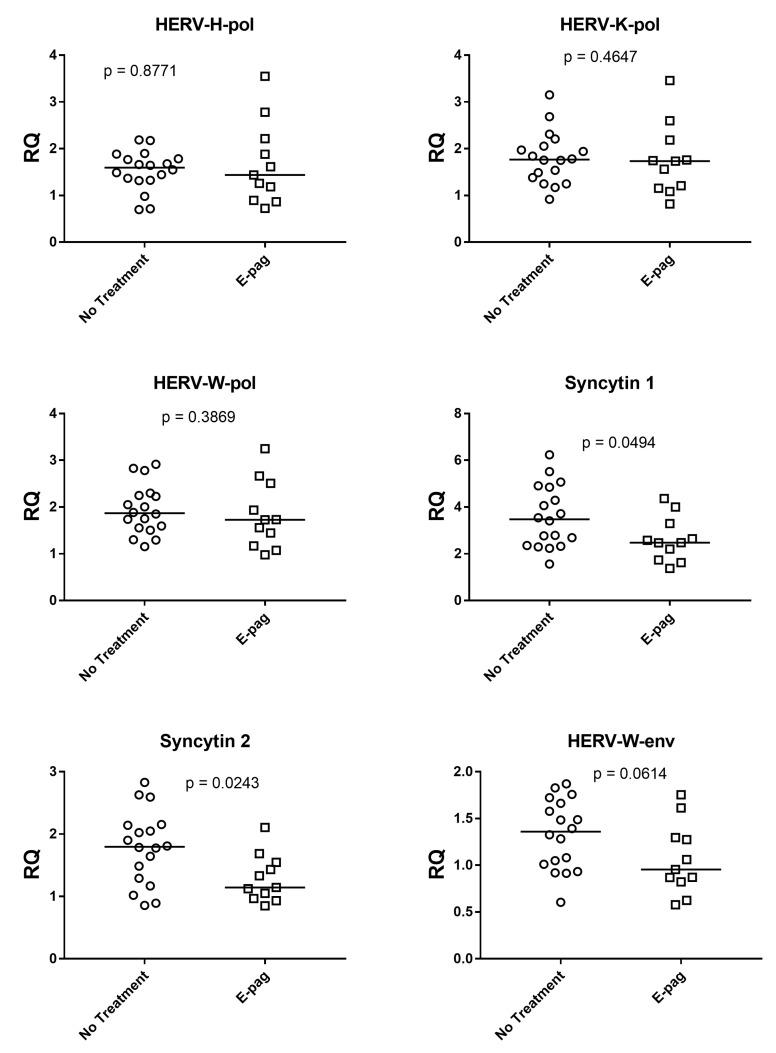
Transcription levels of pol genes of HERV-H, HERV-K, and HERV-W as well as of env genes of Syncytin 1, Syncytin 2, and HERV-W in whole blood from 11 patients on eltrombopag treatment alone (E-pag) and 18 patients with no treatment. RQ: Relative Quantification. Circles and squares show the median of three individual measurements, horizontal lines are the median values. Statistical analysis: Mann–Whitney test was used to compare the transcriptional levels of each target gene.

**Figure 3 genes-14-01569-f003:**
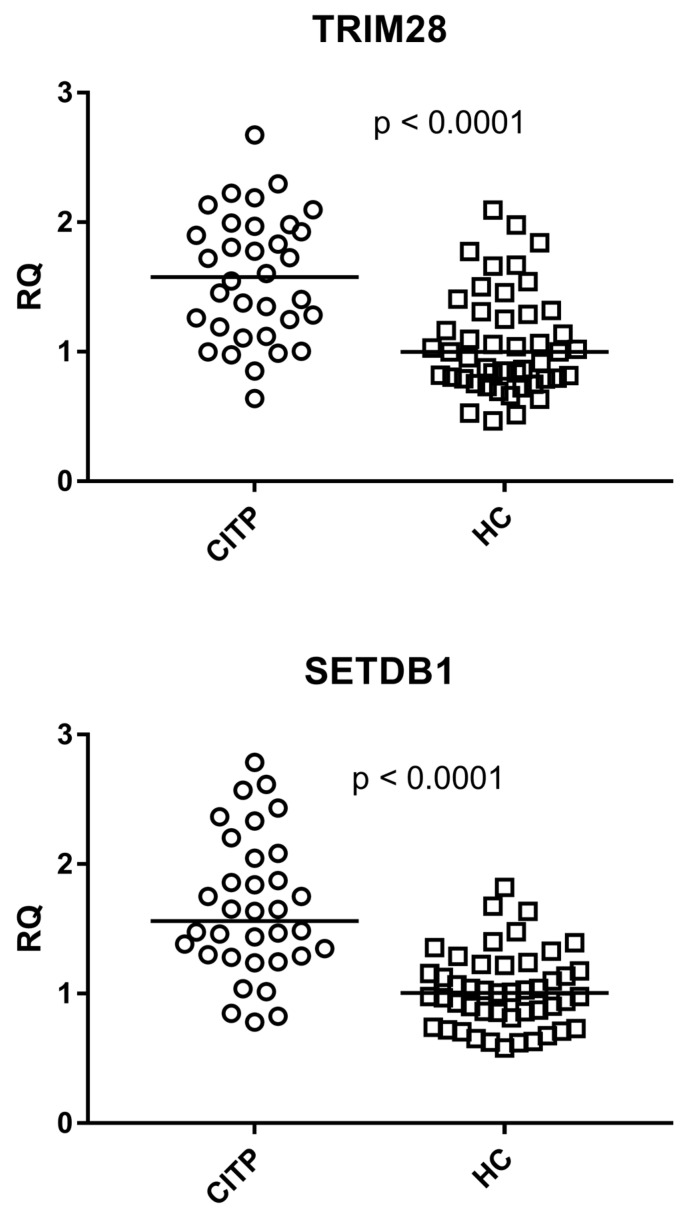
Expression of TRIM28 and SETDB1 in whole blood from 34 patients with chronic immune thrombocytopenia (CITP) and 47 healthy controls (HC). RQ: Relative Quantification. Circles and squares show the median of three individual measurements, horizontal lines are the median values. Statistical analysis: Mann–Whitney test was used to compare the transcriptional levels of each target gene between children with CITP and control children.

**Figure 4 genes-14-01569-f004:**
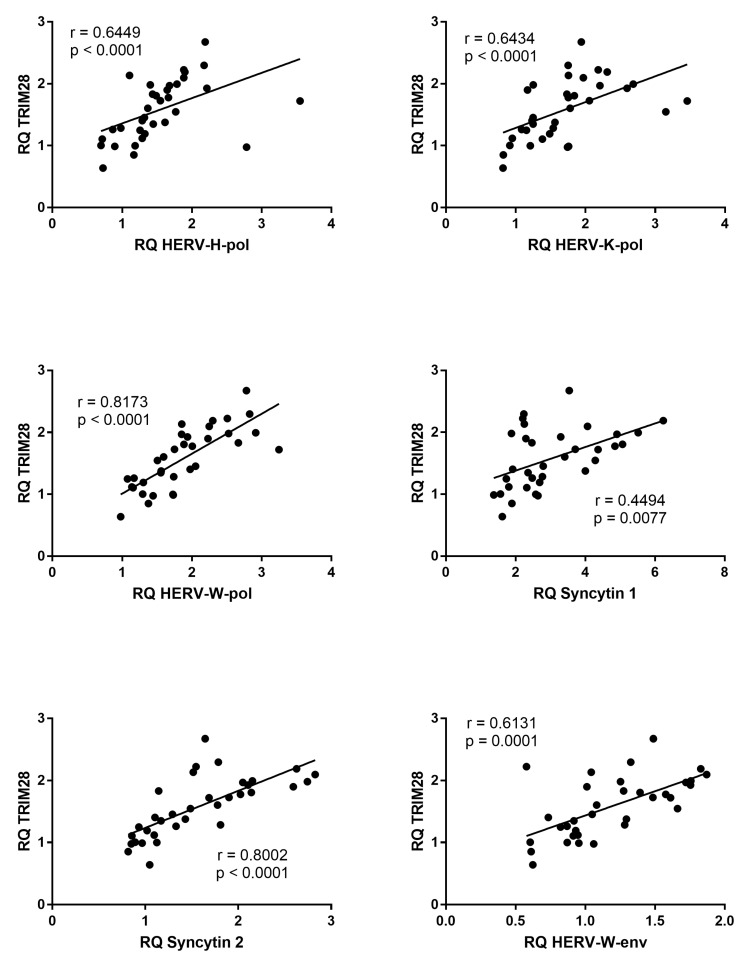
Correlations between transcription levels of TRIM28 and HERV sequences in whole blood from 34 patients with chronic immune thrombocytopenia. RQ: Relative Quantification. Circles show the mean of three individual measurements. Line: Linear regression line. Statistical analysis: Spearman correlation test.

**Figure 5 genes-14-01569-f005:**
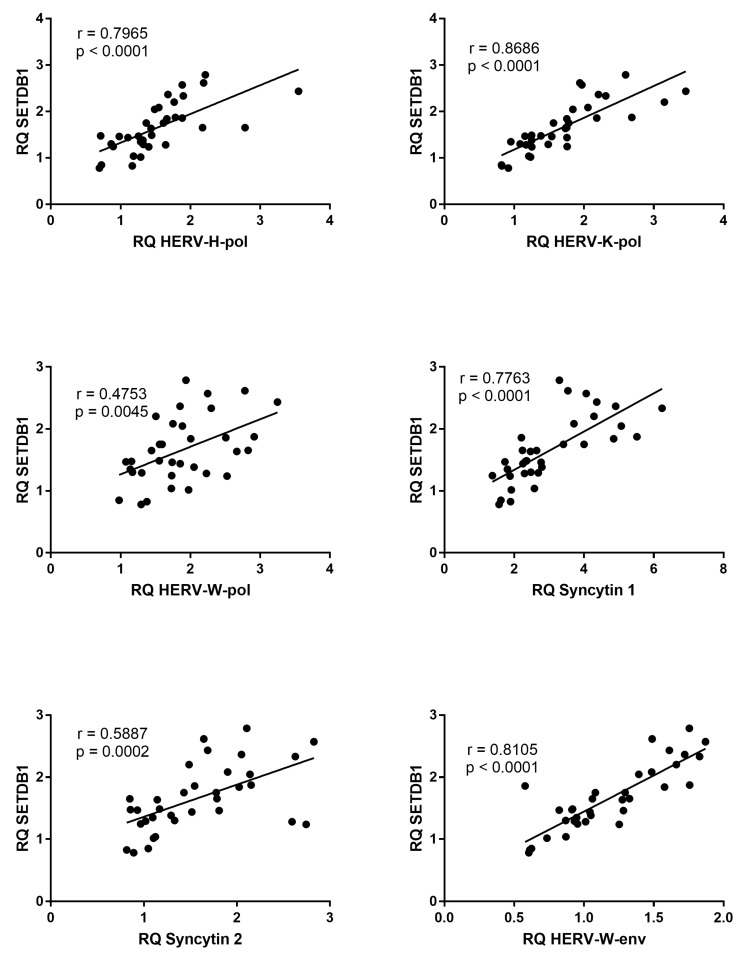
Correlations between transcription levels of SETDB1 and HERV sequences in whole blood from 34 patients with chronic immune thrombocytopenia. RQ: Relative Quantification. Circles show the mean of three individual measurements. Line: Linear regression line. Statistical analysis: Spearman correlation test.

**Table 1 genes-14-01569-t001:** Primers and probes used to assess the transcription levels of pol genes of HERV-K, -W, and –H; env genes of Syncytin 1, Syncytin 2, and HERV-W; TRIM28 and SETDB1; and GAPDH.

Name	Primer/ Probe	Sequence
*HERV-H pol*	Forward	5′-TGGACTGTGCTGCCGCAA-3′
	Reverse	5′-GAAGSTCATCAATATATTGAATAAGGTGAGA-3′
	Probe	6FAM-5′-TTCAGGGACAGCCCTCGTTACTTCAGCCAAGCTC-3′-TAMRA
HERV-K *pol*	Forward	5′-CCACTGTAGAGCCTCCTAAACCC-3′
	Reverse	5′-TTGGTAGCGGCCACTGATTT-3′
	Probe	6FAM-5′-CCCACACCGGTTTTTCTGTTTTCCAAGTTAA-3′-TAMRA
HERV-W *pol*	Forward	5′-ACMTGGAYKRTYTTRCCCCAA-3′
	Reverse	5′-GTAAATCATCCACMTAYYGAAGGAYMA-3′
	Probe	6FAM-5′-TYAGGGATAGCCCYCATCTRTTTGGYCAGGCA-3′-TAMRA
Syncytin 1 *env*	Forward	5′-ACTTTGTCTCTTCCAGAATCG-3′
	Reverse	5′-GCGGTAGATCTTAGTCTTGG-3′
	Probe	6FAM-5′-TGCATCTTGGGCTCCAT-3′-TAMRA
Syncytin 2 *env*	Forward	5′-GCCTGCAAATAGTCTTCTTT-3′
	Reverse	5′-ATAGGGGCTATTCCCATTAG-3′
	Probe	6FAM- 5′-TGATATCCGCCAGAAACCTCCC-3′-TAMRA
HERV-W *env*	Forward	5′-CTTCCAGAATTGAAGCTGTAAAGC-3′
	Reverse	5′-GGGTTGTGCAGTTGAGATTTCC-3′
	Probe	6FAM-5′-TTCTTCAAATGGAGCCCCAGATGCAG-3′-TAMRA
TRIM28	Forward	5′-GCCTCTGTGTGAGACCTGTGTAGA-3′
	Reverse	5′-CCAGTAGAGCGCACAGTATGGT-3′
	Probe	6FAM-5′-CGCACCAGCGGGTGAAGTACACC-3′-TAMRA
SETDB1	Forward	5′-GCCGTGACTTCATAGAGGAGTATGT-3′
	Reverse	5′-GCTGGCCACTCTTGAGCAGTA-3′
	Probe	6FAM-5′-TGCCTACCCCAACCGCCCCAT-3′-TAMRA
GAPDH	Forward	5′-CGAGATCCCTCCAAAATCAA-3′
	Reverse	5′-TTCACACCCATGACGAACAT-3′
	Probe	6FAM-5′-TCCAACGCAAAGCAATACATGAAC-3′-TAMRA

**Table 2 genes-14-01569-t002:** Clinical characteristics of children with chronic immune thrombocytopenia (CITP) and age-matched healthy controls (B1 and B2).

	CITP (*n* = 34)	B1 (*n* = 64)	B2 (*n* = 47)
Mean age (±SD)	12.1 (±4.1) years	10.9 (±3.5) years	10.6 (±3.2) years
Males (%)	17 (50)	35 (55.7)	29 (61.7)
Mean duration of CITP (±SD) at time of sampling	4.4 (±3.1) years		
Ongoing therapy n (%)	Eltrombopag: 11 (31.4) MMF: 2 (5.7) Eltrombopag + MMF: 2 (5.7) Sirolimus: 1 (2.9)		

*n*: number; SD: standard deviation; %: percentage; MMF: mofetil mycophenolate.

## Data Availability

Not applicable.

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
