# Peer review of "Children with Chronic Immune Thrombocytopenia Exhibit High Expression of Human Endogenous Retroviruses TRIM28 and SETDB1"

_genes, 2023, doi:10.3390/genes14081569_

Round 1

Reviewer 1 Report

Tovo et al., present clinically relevant findings whereby the levels of endogenous retroviral (ERV) genes are quantified to be significantly higher in patients with chronic immune thrombocytopenia (CITP) when compared to age-matched healthy controls. This was also correlated to levels of TRIM28/SETDB1, which are known inhibitors of ERV levels, although causal relationship cannot be determined at this stage. This work is very encouraging to provide a better understanding of this disease.

Minor points:

·         The authors should be careful when implications are made between these correlative findings and the disease causality. There are multiple instances in the text, however for example, in the introduction on Page 2 Line 78/79 “…leading to an escalating disease, such as in ITP.” Alternative phrasing could include “reminiscent of the clinical presentation of ITP” or similar.

·         The methods used in this study cannot distinguish the origin of the ERV mRNA, however, can the authors speculate to this (i.e. extracellular/intracellular and/or cell types involved) or point to future work for deciphering this.

·         The authors should include a table with the values of each mRNA quantification for individual patients. This would address questions like: Are the low expressors of HERV-H the same patients as those in HERV-K, HERV-W, etc? Or are they randomly distributed?

·         Can the patients be provided with a clinical score for severity of disease and does this correlate with mRNA levels? Can this be added to the table above?

·         What is the scientific rationale for the Eltrombopag resulting in reduced ERV mRNA levels for Syncytin but not HERV-H, K or W? Treatment status should be included in the table mentioned above.

·         Please include TRIM28 and SETDB1 relative levels in the table mentioned above.

·         Obviously one cannot evaluate everything, but can the authors provide an additional statement in the text indicating why they chose to evaluate TRIM28 and SETDB1 but exclude the plethora of other genes involved in the epigenetic and post-transcriptional regulation of ERVs?

Language related points:

·         Text says Syncytin, graphs say Syncitine

·         Occasionally CTIP is used as an acronym, whereas it should be CITP

Author Response

Tovo et al., present clinically relevant findings whereby the levels of endogenous retroviral (ERV) genes are quantified to be significantly higher in patients with chronic immune thrombocytopenia (CITP) when compared to age-matched healthy controls. This was also correlated to levels of TRIM28/SETDB1, which are known inhibitors of ERV levels, although causal relationship cannot be determined at this stage. This work is very encouraging to provide a better understanding of this disease.

 We appreciate very much the referee’s interest in our study and her/his useful comments.

Minor points:

  • The authors should be careful when implications are made between these correlative findings and the disease causality. There are multiple instances in the text, however for example, in the introduction on Page 2 Line 78/79 “…leading to an escalating disease, such as in ITP.” Alternative phrasing could include “reminiscent of the clinical presentation of ITP” or similar.

       We followed this suggestion, on Page 2 lines 78/79 (of the copy including the changes we made) we used “reminiscent of the clinical presentation of ITP” instead of “leading to an escalating disease, such as in ITP”.        

          The methods used in this study cannot distinguish the origin of the ERV mRNA, however, can the authors speculate to this (i.e. extracellular/intracellular and/or cell types involved) or point to future work for deciphering this.

       We totally agree with your stimulating doubts. To this purpose we are performing a specific study in healthy adults. Preliminary results show that the HERV mRNA levels are at limit of detection in plasma. Among white blood cells they are commonly present in lymphocytes (both T and B cells), but, unexpectedly, their maximum levels are detected in neutrophils.   

  • The authors should include a table with the values of each mRNA quantification for individual patients. This would address questions like: Are the low expressors of HERV-H the same patients as those in HERV-K, HERV-W, etc? Or are they randomly distributed?

We followed this suggestion and included such a table. In particular, we have added on lines 198/199:  “The values of each HERV RNA quantification for individual patients are reported in Supplementary Table S1.

  • Can the patients be provided with a clinical score for severity of disease and does this correlate with mRNA levels? Can this be added to the table above?

       Unfortunately,  the relative low number of total  patients and the administration of drugs in more severe cases did not allow the analysis of the correlation between mRNA levels and disease severity.

  • What is the scientific rationale for the Eltrombopag resulting in reduced ERV mRNA levels for Syncytin but not HERV-H, K or W? Treatment status should be included in the table mentioned above.

      Targeted studies are needed to clarify the underlying biological mechanisms leading to the impaired HERV mRNA levels in treated patients. At present, only mere speculations (with a lot of imagination) could be made.

      Treatment status has been included in the Supplementary Table S1.

  • Please include TRIM28 and SETDB1 relative levels in the table mentioned above.
  1. OK. We have added on lines 236/237: “The mRNA values of TRIM28 and SETDB1 for individual patients are reported in Supplementary Table S1.”

         Obviously one cannot evaluate everything, but can the authors provide an additional statement in the text indicating why they chose to evaluate TRIM28 and SETDB1 but exclude the plethora of other genes involved in the epigenetic and post-transcriptional regulation of ERVs?

      Among the number of cellular genes that can influence the expression of endogenous retroviruses, we underlined in Introduction (lines 85-86) that. “…. Both TRIM28 and SETBD1 play key roles for epigenetic transcriptional modulation of retroviral sequences [38,39].

       Furthermore, we also pointed out in Introduction (lines 86-88) that: ”…. accumulating data demonstrate their direct involvement in cellular homeostasis and in the control of both innate and adaptive immune responses [40,41]. and in Discussion (lines  352-357): “….They influence B lineage differentiation and T cell lineage commitment and activation [73,74]. In particular, through the complex with KRAB-ZFPs and forkhead box P3 (Foxp3), TRIM28/SETDB1 control the expansion of T cells into regulatory phenotypes [40,73], conditioning the Treg suppressor activity or distribution of DCs and T cell priming toward inflammatory effector cells [73,75,76]. Notably, Foxp3 gene polymorphisms are also candidate risk factors in autoimmune diseases, including CITP [10,77]. Therefore, the evaluation of TRIM28 and SETDB1 provides an additional, targeted information in an autoimmune disease such as CITP.

      Finally, the expressions of TRIM28 and SETDB1 were evaluated in other diseases characterized by aberrant expression of HERVs (e.g. references 34,48). Therefore, one can compare the present findings with those emerged in other settings with dysregulation of HERV transcription.      

Language related points:

  • Text says Syncytin, graphs say Syncitine

       We have corrected the graphs. Thanks a lot!

  • Occasionally CTIP is used as an acronym, whereas it should be CITP.

       We have corrected this mistake (line 225). Once again we thank you very much for your useful comments.

Reviewer 2 Report

This manuscript looks at potential epigenetic associations with cITP. In particular the authors examine expression levels of HERVs to determine if these are present in higher concentrations in children and YAs with cITP. The paper is very well written and the data is presented in a clear and well-developed way.

I do have some concerns, however. especially with regards to the associations with eltrombopag. Specifically, the author's compare the levels of HERVs and epigenetic modulators in a small sample of cITP patients who are either on eltrombopag or "no treatment". First, I would clarify what is meant by "no treatment". Does this include no steroids and no IVIg? If not, these confounders need to be considered in the analysis. Similarly, given the small numbers of patients examined in the cohort it would be helpful to compare the cohort to healthy controls as well especially since the patients on eltrombopag may represent a more refractory population requiring additional treatment (if the control arm is truly "no treatment"). I would recommend further clarification of these findings or removing the eltrombopag data (which can be presented in a separate manuscript later once these issues are better characterized). 

Author Response

REVIEWER 2

We appreciate very much the referee’s interest in our study and her/his useful comments.

I do have some concerns, however. especially with regards to the associations with eltrombopag. Specifically, the author's compare the levels of HERVs and epigenetic modulators in a small sample of cITP patients who are either on eltrombopag or "no treatment". First, I would clarify what is meant by "no treatment". Does this include no steroids and no IVIg? If not, these confounders need to be considered in the analysis. Similarly, given the small numbers of patients examined in the cohort it would be helpful to compare the cohort to healthy controls as well especially since the patients on eltrombopag may represent a more refractory population requiring additional treatment (if the control arm is truly "no treatment"). I would recommend further clarification of these findings or removing the eltrombopag data (which can be presented in a separate manuscript later once these issues are better characterized). 

We have clarified thatAs illustrated in Figure 2, the expressions of SYN1 and SYN1 were inhibited significantly, with borderline p value for HERV-W-env, in the 11 patients on Eltrombopag treatment alone as compared to the 18 patients without any treatment (including no administration of steroids or IVIgG)” (Lines 203-204).

Furthermore, following your suggestion, we have  added (lines 204-205): ” When expressions of  HERV-env genes from patients on Eltrombopag alone were compared to those from HC, only SYN1 mRNA levels showed significantly higher values (Supplementary Figure S1).”and in lines 205-207:  “No differences were found for HERV-pol genes between patients on Eltrombopag treatment alone and those with no therapy.